# The Sustainability of Sweet Potato Residues from Starch Processing By-Products: Preparation with *Lacticaseibacillus rhamnosus* and *Pediococcus pentosaceus*, Characterization, and Application

**DOI:** 10.3390/foods12010128

**Published:** 2022-12-27

**Authors:** Lili Zhu, Hongnan Sun, Mengmei Ma, Taihua Mu, Guohua Zhao, Moe Moe Lwin

**Affiliations:** 1Key Laboratory of Agro-Products Processing, Ministry of Agriculture and Rural Affairs, Institute of Food Science and Technology, Chinese Academy of Agricultural Sciences, Beijing 100193, China; 2College of Food Science, Southwest University, Chongqing 400715, China; 3Department of Biotechnology, Mandalay Technnological University, Mandalay 999091, Myanmar

**Keywords:** sweet potato residues, fermentation, nutritional–functional composition, structure, in vitro saliva–gastrointestinal digestion, colonic fermentation

## Abstract

The effects of *Lacticaseibacillus rhamnosus* and *Pediococcus pentosaceus* on the nutritional–functional composition, structure, in vitro saliva–gastrointestinal digestion, and colonic fermentation behaviors of fermented sweet potato residues (FSPR) were investigated. The FSPR was obtained under the condition of a solid-to-liquid ratio of 1/10, inoculation quantity of 1.5%, mixed bacteria ratio 1:1, fermentation time of 48 h, and fermentation temperature of 37 °C. The FSPR showed higher contents of soluble dietary fiber (15.02 g/100 g), total polyphenols content (95.74 mg/100 g), lactic acid (58.01 mg/g), acetic acid (1.66 mg/g), volatile acids (34.26%), and antioxidant activities. As exhibited by FTIR and SEM, the higher peak intensity at 1741 cm^−1^ and looser structure were observed in FSPR. Further, the FSPR group at colonic fermentation time of 48 h showed higher content of acetic acid (1366.88 µg/mL), propionic acid (40.98 µg/mL), and butyric acid (22.71 µg/mL), which were the metabolites produced by gut microbiota using dietary fiber. Meanwhile, the abundance of *Bifidobacterium* and *Lacticaseibacillus* in the FSPR group was also improved. These results indicated that FSPR potentially developed functional foods that contributed to colonic health.

## 1. Introduction

Sweet potato (*Ipomoea batatas* Lam.), an annual or perennial vine herbaceous plant belonging to the Convolvulaceae family, is the fifth principal food crop, following rice, wheat, maize, and potato. The yield of fresh sweet potatoes in China is 49.20 million tons, accounting for 54.97% of the world’s production (FAO, 2020). Sweet potato is commonly used in the process of starch and thereof products; approximately 4.5–5.0 tons of fresh sweet potato residues (SPR) are generated for 1 ton of starch produced [1]. However, only a small amount of SPR is used as low-value animal feed; most of it is discarded directly, leading to serious resource waste and environmental pollution. Due to the rise of public awareness of high value-added utilization for by-products resources in recent decades, there is an increasing study of the extraction of high-value components from by-products, among which the extraction of dietary fiber and pectin from SPR by acids, alkalines, and enzymes are the most in-demand products [2,3]. However, the SPR have not been fully exploited; moreover, waste solutions are unavoidably generated during the production of dietary fiber and pectin, leading to secondary pollution. Thus, an efficient and eco-friendly processing technology is desired to make full use of SPR without any waste production.

Lactic acid bacteria (LAB) fermentation, a traditional and safe biotechnology in the food industry, plays an important role in improving food quality via enhancing the nutritional value, promoting antioxidants, hypolipidemic, and bettering flavor, etc. [4,5]. Recently, LAB has been widely applied in the fermentation of plant-based material, such as blueberry and blackberry juices [5], blueberry pomace [4], jujube juice [6], etc., and different metabolic mechanisms exhibited in different strains. For instance, four kinds of LAB (*L. paracasei*, *L. mesenteroides*, *L. rhamnosus GG*, and *L. plantarum*) and their pairwise complex had been used to ferment tofu whey beverage, and *L. paracasei* combined with *L. rhamnosus* GG fermentation improved the content of isoflavone aglycones and viable cell counts compared to single strain [7]. Furthermore, Bujna et al. [8] found that fermented apricot juice with pairwise complex strains of *Bifidobacterium lactis* Bb-12, *Bifidobacterium longum* Bb-46, *L. casei* 01, and *L.* La-5 exhibited higher viable cell counts and acetic acid contents than those of monoculture, indicating that fermentation with mixed strains showed better fermentation effects than single bacteria. Additionally, Yan et al. [9] reported that rice buckwheat fermented with *Bacillus* DU-106 and *L. plantarum* could better promote the growth of short-chain fatty acids (SCFAs)-producing bacteria, such as *Lacticaseibacillus* and *Blautia*, than unfermented rice buckwheat, to improve the gut microbiota structure. Moreover, Likotrafiti et al. [10] have studied the effects of two kinds of probiotics/prebiotic complexes (*Bifidobacterium*/isomalto-oligosaccharide; *Lacticaseibacillus* /fructose oligosaccharide) on the gut microbiota of the elderly; the results showed that probiotics/prebiotic complexes could significantly promote the growth of *Bifidobacterium* and *Lacticaseibacillus* while inhibiting the growth of *Bacteroides*.

In our previous study, SPR fermented with *Lacticaseibacillus rhamnosus* displayed better effects on improving nutrition, functional properties, and flavors, followed by *Pediococcus pentosaceus* [1]. However, the optimum process conditions of SPR fermented by mixed strains have not been investigated, as well as the nutritional–functional composition, structure, antioxidant activity, sensory characteristics, and probiotic activities. 

The objective of the present study is to explore the optimum process condition of SPR fermented by mixed *Lacticaseibacillus rhamnosus* and *Pediococcus pentosaceus*, as well as the nutritional–functional composition, structure, antioxidant activity, sensory characteristics, in vitro saliva–gastrointestinal digestion, and colonic fermentation behaviors of FSPR. These findings could provide a novel, environment-friendly processing technology and theoretical basis for the comprehensive utilization of SPR and other by-products from fruits, vegetables, and grains processing to develop functional foods with zero waste generation. 

## 2. Materials and Methods

### 2.1. Chemicals and Materials

Sweet potato residues obtained from starch processing were provided by Shandong Huaqiang Agricultural Science and Technology Development Co., Ltd. (Linyi, China). *Lacticaseibacillus rhamnosus* CICC 23119 (LR) and *Pediococcus pentosaceus* CICC 21862 (PP) used in this study were purchased from China Center of Industrial Culture Collection (Beijing, China).

The total starch assay kit was purchased from Megazyme International Ireland (Bray Business Park, Bray, Co. Wicklow, Ireland). Pancreatin (4000 U/g), pepsin (3000 U/g), lactic acid, acetic acid, propionic acid, and butyric acid were purchased from Sigma-Aldrich (St. Louis, MO, USA). α-amylase (≥3000 units/mL), protease (≥3000 units/mL) and amyloglucosidase (≥5000 units/mL), bile salt, fructo-oligosaccharides (FOS), tween 80, and all other chemicals and reagents were analytical grade and purchased from Beijing Solarbio Science & Technology Co., Ltd. (Beijing, China).

### 2.2. Investigation of Optimum Process Conditions of Fermented SPR 

LR and PP were cultured in Man–Rogosa–Sharpe (MRS) medium at 37 °C for 36 h according to the methods provided by China Center of Industrial Culture Collection (Beijing, China). Then the bacterial suspensions were washed and diluted to 10^8^ cfu/mL with phosphate-buffered saline (PBS) solution for SPR fermentation, and cell counts on plates and OD_600_ value were combined to check its viability [11].

A design of single-factor experiment approach was applied to optimize the fermentation conditions. In brief, the SPR were immersed in distilled water at different ratios (1:5, 1:10, 1:15, 1:20, and 1:40, *w*/*v*), 0.5% (*v*/*v*) thermostable α-amylase solution was added, and hydrolyzed at 95 °C for 30 min, after cooling to room temperature, the pH was adjusted to 6.80 with 4 M NaOH. Subsequently, sterilized at 121 °C for 15 min, after cooling, different inoculation quantities (0.5%, 1.5%, 3.0%, 4.5%, and 6.0%, *v*/*v*) of different mixed bacteria ratios (1/3, 1/2, 1/1, 2/1, and 3/1, LR/PP) were allotted and then incubated at different temperatures (31, 34, 37, 40, and 43 °C) for different fermentation times (24, 36, 48, 60, and 72 h). The organic acid contents and viable bacteria count were adopted to evaluate the fermentation efficiency affected by above five main factors. The fermented SPR under optimal conditions was named FSPR for further analysis.

### 2.3. Analysis of Nutritional Composition

Protein, ash, moisture, starch, total dietary fiber, soluble dietary fiber, and insoluble dietary fiber were assessed according to AOAC Official Method. Viable bacteria count was determined by cell counts on plates method described by the study of Duedu et al. [11].

The reducing sugars were measured by 3, 5-dinitrosalicyclic acid method [1]. In brief, samples were extracted by distilled water at 50 °C for 20 min at a shaking speed of 60 rpm three times, then centrifuged at 4000× *g* for 5 min, and the supernatants were collected for determination. The glucose was used as standard, and absorbance was read at 540 nm by a UV1101 spectrophotometer (Hitachi, Tokyo, Japan).

The soluble protein was determined by Lowry method [1]. Briefly, samples were mixed with distilled water and extracted under ultrasound for 10 min at room temperature, then centrifuged at 4000× *g* for 15 min, and the supernatant was for analysis. Bovine serum albumin (BSA) solutions at concentration of 1–100 mg/mL were used to make standard curve, and the absorbance was measured at 750 nm. Additionally, total and free amino acids were analyzed by Automatic Amino Acid Analyzer (L-8900, Hitachi, Japan) according to our previous study [1]. 

### 2.4. Total Polyphenols Content (TPC) and Antioxidant Activities

The TPC were extracted with 70% ethanol (*v*/*v*) and measured by Folin–Ciocalteu method [1]. Gallic acid standards at 0.02 to 0.10 mg/mL were used to make calibration curve, and the TPC was expressed in milligram of gallic acid equivalent per 100 g of dry weight (mg GAE/100 g, DW).

The antioxidant activities were performed as described by our previous study [1]. For ABTS radical scavenging capacity analysis, 7 mM ABTS mixed with 2.45 mM potassium persulfate at a ratio of 1:1 (*v*/*v*) and incubated at room temperature in the dark for 12–16 h to prepare ABTS stock solution; the stock solution was diluted with 70% ethanol until the absorbance was 0.70 at 734 nm. Then, 0.15 mL sample was reacted with 2 mL diluted ABTS solution at room temperature for 6 min, and the absorbance was read at 734 nm.

For DPPH radical scavenging activity analysis; 0.15 mL sample and 3.0 mL of DPPH solution (0.2 mM in 70% ethanol) were mixed and incubated at room temperature in dark for 30 min; the absorbance was read at 517 nm. The ascorbic acid standard curve was prepared at concentrations ranging from 0 to 50 μg/mL, and ABTS radical scavenging capacity and DPPH radical scavenging activity were expressed as mg ascorbic acid equivalents (AAE)/100 g DW.

For FRAP capacity analysis, 0.15 mL sample and 2.85 mL of FRAP solution were mixed and incubated at 37 °C in dark for 30 min, and the absorbance was recorded at 593 nm. Trolox solutions at concentrations ranging from 10 to 200 μg/mL were used to make standard curve, and the FRAP activity was expressed as mg Trolox equivalents (TE)/100 g DW.

### 2.5. pH Value, Total Titratable Acid (TTA) and Organic Acids

The pH value was measured by a pH meter (PHS-25, Shanghai Precision Scientific Instruments Company, Shanghai, China). The TTA was expressed in mL NaOH/g DW. Sample solutions were titrated by 0.1 M NaOH, and phenolphthalein solution (0.1%, m/v) was used as endpoint indicator [12].

The concentration of lactic acid and SCFAs were determined by high-performance anion exchange chromatography-pulsed amperometric detector (HPAEC-PAD) (HPAEC, Model 1200, Agilent, Palo Alto, CA, USA) [1]. The fermentation liquids were centrifuged at 4000× *g* for 10 min and filtered with a 0.22 μm filter membrane, and 25 μL of the filtrate was injected into the chromatography. An AS11-HC (4 × 250 mm) column held at 30 °C, 0.8 mmol/L KOH solution as a mobile phase at a flow rate of 1.0 mL/min was used for the separation of acids.

### 2.6. GC-MS Analysis

According to the methods of our previous study [1]. The volatile compounds of samples were collected and analyzed by solid-phase micro extraction (SPME) and GC–MS (QP 2010 plus, Shimadzu Corp, Kyoto, Japan) equipped with a DB-WAX column (30 m × 0.25 mm × 0.25 μm), respectively.

### 2.7. Microstructure Analysis

#### 2.7.1. Fourier-Transform Infrared (FTIR) Spectroscopy Analysis

The FTIR spectra were determined by a Tensor 27 spectrometer (Bruker, Karlsruhe, Germany) according to the study of Ma et al. [13]. Briefly, 2 mg sample powder was thoroughly mixed with 200 mg KBr (*w*/*w*) and condensed into pellets. Then the FTIR spectra were recorded in the frequency range from 4000 to 400 cm^−1^ at a resolution of 4 cm^−1^ against KBr as the background.

#### 2.7.2. Scanning Electron Microscopy (SEM) Analysis

The microstructure was observed using SEM (SU 8010, Hitachi, Tokyo, Japan), with magnification at 1000× and voltage at 10 kV, according to the report of Ma et al. [13]. The samples were fixed and sputter coated using an ion sputter coater (MC1000, Hitachi, Tokyo, Japan).

### 2.8. In Vitro Saliva–Gastrointestinal Digestion and Colonic Fermentation

#### 2.8.1. In Vitro Saliva–Gastrointestinal Digestion

The in vitro digestion of samples was evaluated according to the report of Feng et al. [14] with slight modifications. Samples of 1 g were mixed with 20 mL distilled water and balanced at 37 °C for 10 min. Then, 0.05 mL 0.1% simulated salivary juice (0.1% α-amylase solution in distilled water, *w*/*v*) was added and reacted at 37 °C in a water bath shaker for 1 min (100 rpm). After salivary digestion, the pH was adjusted to 2.0 with 6 M HCl, then 1 mL simulated gastric juice (2% pepsin in 0.1 M HCl, *w*/*v*) was added and incubated at 37 °C for 1 h (100 rpm). Next, the pH value of the mixture was adjusted to 7.2 with 1 M NaHCO_3_, 5 mL simulated intestinal juice (1.2 g bile salt and 0.2 g trypsin in 100 mL of 0.1 M NaHCO_3_), and 5 mL sodium chloride/potassium chloride mixed solution (7.02 g/L NaCl, 0.37 g/L KCl) were added and incubated at 37 °C for 2 h (100 rpm). A part of samples was collected to determine viable bacteria counts and TPC. Another part was exposed to boiled water to inactivate the enzyme and freeze-dried for the extraction of total dietary fiber (TDF).

#### 2.8.2. In Vitro Colonic Fermentation

In vitro colonic fermentation of samples by fresh human feces was carried out according to the method of Zhou et al. [15] with minor modifications. Fresh fecal samples were collected from three healthy volunteers (two females and one male, aged 22 to 26) who did not have digestive diseases or any treatment of antibiotics in the past three months. The fecal samples were diluted with sterilized PBS solution (0.1 M, pH 7.2) to obtain 10% of fecal slurry (*w*/*v*) with a magnetic mixture of 10 min, thereafter filtered, pooled, and strained through 6 layers of cheesecloth to obtain fecal inoculum. Further procedures were performed within 1 h after collection. The basal nutrient medium was composed of yeast extract (4.5 g), peptone (3.0 g), tryptone (3.0 g), NaCl (4.5 g), KCl (2.5 g), K_2_HPO_4_ (0.04 g), KH_2_PO_4_ (0.04 g), MgSO_4_ (0.01 g), CaCl_2_ (0.01 g), NaHCO_3_ (2.0 g), bile salt (0.5 g), L-cysteine salt (0.5 g), hemin (0.05 g), tween 80 (2.0 mL), distilled water (1.0 L), and the pH of basal nutrient medium was adjusted to 7.0.

Accurate 60 mg of samples and 4.5 mL fecal inoculum were added to the 5.5 mL basal nutrient medium. The culture medium without sample was set as the blank control (CK). The mixture was incubated in an anaerobic atmosphere of 10% H_2_, 10% CO_2_, and 80% N_2_ at 37 °C. Samples were collected after 12, 24, and 24 h of fermentation, respectively, and stored at −80 °C for subsequent experiments. 

#### 2.8.3. Gut Microbiota Analysis

The total microbial DNA of fecal ferments was extracted, quantified, and checked by OMEGA-soil DNA Kit (Omega Bio-Tek, Norcross, GA, USA), NanoDrop 2000 UV–vis spectrophotometer (Thermo Scientific, Wilmington, NC, USA), and 1% agarose gel electrophoresis, respectively. The amplification of the bacterial 16S rRNA genes of V3–V4 region was performed by primers 338F (5′-ACTCCTACGGGAGGCAGCAG-3′) and 806R (5′-GGACTACHVGGGTWTCTAAT-3′) with PCR system (GeneAmp 9700, ABI, Foster City, CA, USA) according to the standard protocols by Majorbio Bio-Pharm Technology Co., Ltd. (Shanghai, China). Amplification was accomplished under the following reactions: 95 °C for 3 min, followed by 40 cycles at 95 °C for 30 s, 60 °C for 30 s, and 72 °C for 45 s [16]. After the PCR products were extracted, purified, quantified, and sequencing libraries generated, the Illumina MiSeq platform (San Diego, CA, USA) was used for high-throughput sequencing and bioinformatics analysis according to the standard protocols by Majorbio Bio-Pharm Technology Co., Ltd. (Shanghai, China).

#### 2.8.4. Extraction and Characterization of TDF from FSPR during Different In Vitro Stages

The TDF was extracted as described by AOAC 991.43 method, and the small molecule salts in TDF were removed by 3.5 kDa retention membrane dialysis. A Tensor 27 spectrometer was applied to obtain the FTIR spectra. The X-ray diffraction (XRD) patterns were analyzed by X’Pert PRO X-ray diffractometer (PNAalytical, B.V., Almelo, The Netherlands), and the degree of crystallinity was calculated according to the study of Ma et al. [13]. Furthermore, the SEM was employed to observe the microstructure of TDF.

### 2.9. Statistical Analysis

Statistical analyses were performed using SPSS 16.0 (SPSS Inc., Chicago, IL, USA) software. Data were subjected to one-way ANOVA analysis of variance, and Duncan’s multiple range test was used for comparison of group means with statistical significance at *p* ˂ 0.05. All experiments were performed in triplicate with values expressed as the mean ± standard deviation (SD) of the data obtained.

## 3. Results

### 3.1. Effects of Fermentation Parameters on the Organic Acid Contents and Viable Bacteria Count of Fermented SPR

Many previous studies have confirmed that organic acid contents and viable bacteria count represented fermentation efficiency, which were used as indicators for LAB fermentation [7,17]. The effects of the solid-to-liquid ratio, inoculation quantity, mixed bacteria ratio, fermentation temperature, and fermentation time on organic acid contents and viable bacteria count of SPR fermented with combined strains are shown in Figure 1.

As shown in Figure 1A, a, the organic acid contents and viable bacteria count firstly increased and then decreased at the range of 1/5 to 1/20, among which the solid-to-liquid ratio at 1/10 and 1/15, the organic acid contents and viable bacteria count was higher, which were 54.52 mg/g and 55.07 mg/g, respectively, and 9.42 log CFU/g and 9.47 log CFU/g, respectively. Moreover, there was no significant difference between the solid-to-liquid ratio at 1/10 and 1/15. The solid-to-liquid ratio was directly related to the nutrient concentration, which affects the fermentation efficiency and utilization rate of the substrate by influencing the growth and metabolism of microorganisms [18]. Considering the drying costs, a solid-to-liquid ratio of 1/10 was selected. Additionally, the organic acid contents and viable bacteria count were significantly increased in the inoculation quantity range of 0.5–1.5%, and no significant changes were observed in 1.5–6% (Figure 1B,b). A low inoculum amount would prolong the retardation period of microbial growth, and fermentation nutrients cannot meet the needs of microorganisms with high inoculum amounts. This result was similar to the report of Gupta et al. [19], who investigated the oats/sucrose complexes fermented by *L. plantarum* with different inoculum amounts, and the results showed that the viable bacteria count did not continue to increase with the excessive inoculum amounts. Moreover, the organic acid contents and viable bacteria count of mixed bacteria ratio at 1:1 showed no significant difference with 2/1 and 3/1, while it was significantly higher than that of 1/3 and 1/2 (*p* < 0.05) (Figure 1C,c). According to our previous study [1], PP was more favorable for the production of soluble dietary fiber (SDF) than LR, and the mixed bacteria ratio at 1: 1 was selected for its contribution to higher SDF content.

Furthermore, fermentation temperature affects the growth and metabolism of microorganisms [7]. The highest organic acid contents (55.07 mg/g) and viable cell count (9.47 logCFU/g) were observed at 37 °C, indicating lower or higher temperatures were not favorable for LR and PP cultivation (Figure 1D,d). Similar results were observed by Mustafa et al. [20], who investigated the effects of different *Lacticaseibacillus* (*L. plantarum*, *L. casei*, *L. bulgaricus*, and *L. salivarius*) fermentation on the quality of Punica granatum juice, and the highest lactic acid and viable cell count were observed in *L. casei* at 37 °C. Meanwhile, the organic acid contents and viable bacteria count increased within 48 h, which reached 55.07 mg/g and 9.39 log CFU/g, respectively, and no significant increment was observed thereafter (Figure 1E,e), which might be due to the accumulation of organic acids and consumption of nutrients inhibiting the growth and metabolism of microorganisms [7]. Thus, the optimum processing conditions for the production of FSPR were solid-to-liquid ratio 1/10, inoculation quantity 1.5%, mixed bacteria ratio 1:1, fermentation temperature 37 °C, and fermentation time 48 h.

### 3.2. Chemical Characterization of FSPR under Optimal Process Conditions

#### 3.2.1. Nutritional–Functional Compositions

The nutritional–functional compositions of FSPR are shown in Table 1. Compared to SPR (51.01 g/100 g DW), the total starch content of FSPR was decreased to 3.40 g/100 g DW. Meanwhile, the reducing sugar content of SPR (2.50 g/100 g DW) was obviously lower than that of FSPR (18.02 g/100 g DW), which was due to the hydrolysis of starch into small molecular sugars by α-amylase [1]. It is worth noting that the SDF increased from 10.17 to 15.02 g/100 g DW, while insoluble dietary fiber (IDF) decreased from 25.09 to 20.59 after fermentation by LR combined with PP. This might be attributed to the degradation of polysaccharide chains and exposure of more hydrophilic groups caused by heating and enzymatic hydrolysis during the preparation of FSPR [1]. Dietary fibers, especially SDF, exhibit several health benefits, including serum lipid reduction, gut microbiota structure regulation, colon cancer prevention, etc. [3,13]. Additionally, Wu et al. [21] reported that both SDF and IDF exhibited potential prebiotic effects, and SDF could better promote the growth of *Lacticaseibacillus* and *Bifidobacterium* than IDF.

The protein content of FSPR (3.88 g/100 g DW) showed no difference compared to SPR (3.95 g/100 g DW), which was consistent with the results of total and free amino acids (Appendix A). However, the soluble protein content decreased from 475.74 to 141.53 mg/100 g DW after fermentation with combined strains, which might be attributed to the polymerization and precipitation of soluble protein subjected to heating during the preparation of FSPR [1]. Despite the small amount of protein present in FSPR, the protein including peptides and glycoprotein may exert antioxidant activities, anticancer, and suppress lipogenesis biological functions [22].

The ash content of FSPR (3.29 g/100 g DW) was higher than that of SPR (2.41 g/100 g DW), which could be due to the addition of NaOH and PBS solution during the preparation of FSPR. Ash reflects the mineral contents of food, as reported by Ju et al. [23]. Sweet potato residues are abundant in K, Mg, Ca, Fe, etc., which plays a vital role in blood pressure regulation, neuromuscular conduction, secretion of hormones and enzymes, synthesis of heme proteins, etc.

#### 3.2.2. TPC and Antioxidant Activities

TPC and antioxidant activities of FSPR are also presented in Table 1. The TPC was increased from 70.47 to 95.74 mg/100 g DW after fermentation with combined strains. Similar results were found by Vivek et al. [24], whereby the TPC of Sohiong juice was increased after fermentation with *L. plantarum*, which might be due to the hydrolysis of phenolic compounds into simpler forms or/and the release of bounded phenolic compounds from plant cell wall [1]. Correspondingly, the DPPH· scavenging capacity (57.07 mg AAE/100 g DW) and FRAP capacity (60.51 mg TE/100 g DW) of FSPR were higher than that of SPR (33.92 mg AAE/100 g DW, 27.18 mg TE/100 g DW). However, the ABTS· scavenging capacity of FSPR (13.70 mg AAE/100 g DW) showed no difference with SPR (12.70 mg AAE/100 g DW). The higher antioxidant activity of FSPR might be in correlation with its higher TPC contents. Additionally, polyphenol types and molecular structure, especially the number and position of phenolic OH groups, also influence antioxidant activity [1]. Li T. et al. [6] found the DPPH· scavenging activity and FRAP were positively correlated with caffeic acid and rutin, which were increased in jujube juices after fermentation with LAB. Similar results were also observed by Yan et al. [4]; thereby, the antioxidant activity of blueberry pomace was increased after fermentation with *L. rhamnosus* GG and *L. plantarum*-1.

#### 3.2.3. pH Value, TTA, and Organic Acids

The changes in pH value, TTA, and organic acids of FSPR are also displayed in Table 1. After fermentation with combined strains, the pH value was decreased from 5.48 to 3.26, and the TTA increased from 1.06 to 6.86 mL NaOH/g DW. Meanwhile, the lactic acid was greatly increased from 0.05 to 58.01 mg/g DW after fermentation with combined strains, which could enhance flavor and prolong shelf life by giving food a sour taste and inhibiting the growth of pathogens [25]. Furthermore, FSPR showed higher acetic acid (1.66 mg/g DW) than SPR (0.10 mg/g DW), which as one kind of SCFA, could regulate inflammation, liver and cholesterol metabolism, and intestinal barrier function [26]. The above results might be attributed to the production of lactic acid and acetic acid by LAB through hetero-lactic acid metabolism and/or homo-lactic acid metabolism [25]. Similar results were observed by Li H. et al. [27], who reported that the lactic acid of apple juice was significantly increased after fermentation with *L. plantarum*.

#### 3.2.4. GC-MS analysis

Volatile compounds of FSPR are presented in Table 2. The total acid abundance of FSPR (34.26%) was dramatically increased compared to SPR, which could confer the product’s appropriate sour taste. The hexanoic acid and sorbic acid abundance of FSPR were 1.52% and 28.68%, respectively, which exhibited potential inhibiting effects on yeasts and molds [1]. Furthermore, the nonanal and 3-Hydroxybutanone were observed in FSPR with an abundance of 10.98% and 1.59%, respectively, which could give products citrus and flower/cream delightful odor [6]. Additionally, the 2-pentyl-furan abundance of FSPR (12.19%) was higher than that of SPR (10.16%), which might be attributed to the formation of these furan derivatives during the preparation of FSPR [1].

#### 3.2.5. Structural Analysis of FSPR

##### FTIR

The FTIR spectrum of SPR and FSPR is illustrated in Figure 2a. The characteristic peaks at 3394 cm^−1^ attributed to the O-H stretching vibration in hemicellulose and cellulose, and peaks at 2920 cm^−1^ corresponded to the C-H stretching vibration of the methylene group of the polysaccharides, which were observed in both FSPR and SPR [28]. The peaks at 1632 cm^−1^ were ascribed to the aromatic benzene in lignin. The peak intensity and area at 3394 cm^−1^, 2920 cm^−1^, and 1632 cm^−1^ in FSPR were weaker than that of SPR, which was attributed to the decrease in hemicellulose, cellulose, starch, and lignin after fermentation [14]. Interestingly, the peak intensity and area of FSPR at 1741 cm^−1^ were greater than SPR, which was C=O stretching of COOH in uronic acid, indicating an increase in uronic acid and SDF, which was consistent with the increased SDF contents of FSPR in Table 1 [13].

##### SEM

As shown in Figure 2b, SPR showed an obvious starch structure, while no starch granules were observed in FSPR, which was attributed to the hydrolysis of starch by α-amylase. Meanwhile, a similar dietary fiber structure was observed in SPR and FSPR. Moreover, the dietary fiber structure of FSPR was looser than SPR, which might be due to bonds breaking between dietary fibers caused by high-temperature treatment during the preparation of FSPR [29].

### 3.3. Viable Bacteria Count and TPC of FSPR during In Vitro Digestion and Colonic Fermentation

As presented in Figure 3a, the viable bacteria count of FSPR was significantly reduced from 6.87 to 5.86 log CFU/g during gastric digestion (*p* < 0.05), which might be attributed to cell wall damage and viability loss of bacteria subjected to the gastric juice with low pH value [30]. Meanwhile, it further decreased significantly during the intestinal digestion stage to 5.60 log CFU/g (*p* < 0.05). Similar results were also observed by Costa et al. [31], whereby survival of *L. rhamnosus* GG decreased during in vitro gastrointestinal digestion.

As illustrated in Figure 3b, the TPC of FSPR and SPR showed no significant difference during saliva–gastrointestinal digestion, while it increased significantly during colonic fermentation (*p* < 0.05). This was in accordance with the results reported by Dong et al. [32], whereby the TPC of carrot powder increased significantly during colonic fermentation. The increase in TPC during colonic fermentation might be due to the release of polyphenols bound to dietary fiber by the effects of gut microbiota, which could produce carbohydrate hydrolases to decompose cell wall fibers and make the enzymes from the microbiota more accessible to release more of the bound polyphenols [33].

### 3.4. The Value of Total Acids, pH Value, Lactic Acid and SCFAs during Simulated Colonic Fermentation

SCFAs are metabolites of indigestible polysaccharides fermented by specific anaerobic gut microbiota, which play an important role in shaping gut microbiota structure, activating the immune system, regulating host physiology, and energy homeostasis [16]. As depicted in Figure 4, the pH value was decreased while the concentration of SCFAs and total acids were increased with the prolonged colonic fermentation time. Moreover, the FSPR at colonic fermentation time of 48 h had the lowest pH value (6.36), the highest concentration of total acids (1540 µg/mL), acetic acid (1366 µg/mL), and butyric acid (22.71 µg/mL). Meanwhile, the FSPR could better promote the production of SCFAs and total acids compared to SPR and FOS. Acetic acid and propionic acid contribute to liver and cholesterol metabolism regulation, and butyric acid devote to the protection of colonic mucosa, the maintenance of epithelial cell integrity, inhibition of inflammation, and regulation of gene expression in colon cells [26]. It is worth noting that the lactic acid of FSPR was reduced from 112.59 to 104.64 µg/mL, and butyric acid was increased from 18.85 to 22.71 µg/mL during colonic fermentation time of 24 to 48 h, which might be due to the conversion of lactic acid into butyric acid by eubacteria [16]. 

### 3.5. The Effects of FSPR on Gut Microbiota Compositions during Simulated Colonic Fermentation

Stable gut microbiota plays a crucial role in health via resisting pathogenic infections, maintaining immune system balance, and regulating energy metabolism [16,34]. Understanding the relationship between FSPR, gut microbiota, and its metabolites could provide a theoretical basis for preventing disease and promoting health by modulating gut microbiota. The α-diversity defined by community richness (ace and chao) and community diversity (shannon and simpson) of gut microbiota during simulated colonic fermentation are presented in Table 3. The community richness and diversity of the FSPR were decreased in comparison with CK, which might be due to the inability of some pathogenic bacteria to survive exposure to low pH values. Consistent results were found in Figure 5a; the genus number in CK at fermentation time of 12 h, 24 h, and 48 h were 179, 183, and 187, respectively, which were higher than that of FSPR at the same fermentation time. Similar results were also observed by Zhao et al. [16], who reported that the diversity of gut microbiota was decreased with different molecular weights of pectin treatments.

The gut microbiota compositions were characterized in Figure 5b. The abundance of *Bifidobacterium* in FSPR was increased from 3.33 to 4.85% within the fermentation time of 24–48 h. Meanwhile, the abundance of *Bifidobacterium* in FSPR at 48 h was higher than that of CK (3.57%) and FOS (2.41%). *Bifidobacterium* is one of the most important probiotics in the gut, which is beneficial to human health via controlling serum and cholesterol levels, preventing intestinal diseases, regulating the immune system, and inhibiting cancer activity [16,34]. Moreover, the abundance of *Lacticaseibacillus* in FSPR and FOS was increased in comparison with CK, which was regarded as SCFAs producer genera to generate lactic acid and acetic acid [34]. Moreover, the abundance of *Escherichia-Shigella* in FSPR, SPR, and FOS increased compared with CK, and the abundance of *Escherichia-Shigella* obviously decreased in FSPR and SPR while slowly increased in FOS within fermentation time 24–48 h. These results are possibly due to the small molecular sugars in FSPR, and SPR were consumed while still available in FOS during fermentation time of 24–48 h. *Escherichia-Shigella,* as reported, can use low molecular sugars while cannot utilize polysaccharides due to the lack of carbohydrase [35]. *Megamonas* is a characteristic genus of gut microbiota from Asian populations, which has been reported to be higher in colorectal cancer patients [36]. Compared to CK, the abundance of *Megamonas* in FSPR at fermentation time 48 h decreased from 3.39% to 2.95%. The above results indicate that FSPR had a certain effect on improving gut health. Similar results were reported by Tang et al. [37], whereby mulberry pomace fermented by *L. plantarum* could regulate the gut microbiota community by promoting the growth of beneficial bacteria, such as *Lacticaseibacillus* and *Bifidobacterium*.

### 3.6. Characterization of TDF from FSPR during In Vitro Digestion and Colonic Fermentation

#### 3.6.1. FTIR

As illustrated in Figure 6a, the peak intensity at 3394 cm^−1^, 2920 cm^−1^, 1741 cm^−1^, and 1632 cm^−1^ in TDF during colonic fermentation were weaker than that of FSPR, and especially the C48 had the weakest peak intensity. This might be due to the utilization of TDF by gut microbiota, including hemicellulose, cellulose, lignin, and pectin. Wu et al. [21] reported that both IDF and SDF from bamboo shoots could be utilized by gut microbiota to produce SCFAs, and the abundance of *Lacticaseibacillus* and *Bifidobacterium* was increased, which contributed to the effect of regulating gut health.

#### 3.6.2. XRD

The crystallinity was significantly increased after colonic fermentation, of which C48 had the highest crystallinity of 22.12% (Figure 6b), indicating the amorphous region of TDF from FSPR was preferentially utilized by gut microbiota during in vitro colonic fermentation, thereby increasing the ratio of crystalline regions [38]. Additionally, the crystalline structure of dietary fiber could be destroyed by physical treatment, thus increasing the functional properties by making it easier to contact gut microbiota [39].

#### 3.6.3. SEM

As illustrated in Figure 6c, the TDF in simulated colonic fermentation stages exhibited a looser structure, which had obvious dietary fiber breakage with the occurrence of short rod-like and granular materials. This might be attributed to the degradation of dietary fiber by gut microbiota during in vitro colonic fermentation. Deehan et al. [40] reported that dietary fiber could be fermented by gut microbiota to produce SCFAs, which could promote gut health by maintaining the integrity of the intestinal barrier, inhibiting oxidative stress, improving microbiota structure, etc., and the fermentation efficiency was affected by dietary fiber structure and composition.

## 4. Conclusions

In summary, FSPR had higher SDF, TPC, lactic acid, acetic acid, volatile acids, and antioxidant activities. Meanwhile, the TPC was further increased during simulated colonic fermentation. Additionally, the FSPR group showed higher SCFAs contents and *Bifidobacterium* and *Lacticaseibacillus* abundance. This study provides a novel, green and environmental-friendly processing technology for the full use of SPR and demonstrates the potential digestion and fermentation mechanism of FSPR, which indicates that FSPR has the potential to develop functional foods.

## Figures and Tables

**Figure 1 foods-12-00128-f001:**
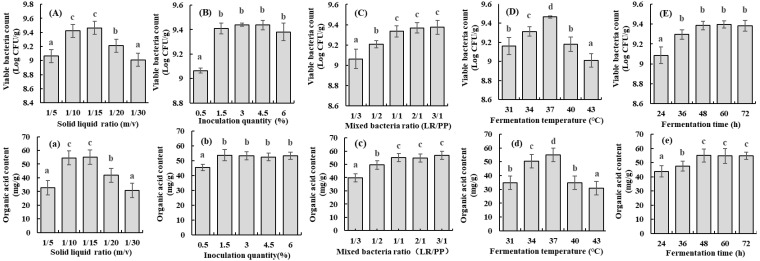
(**A**–**E**): The viable bacteria counts of sweet potato residues fermented with combined strains under different fermentation parameters; (**a**–**e**): The organic acid content of sweet potato residues fermented with combined strains under different fermentation parameters; Lowercase letters above each bar chart indicates significant differences (*p* < 0.05).

**Figure 2 foods-12-00128-f002:**
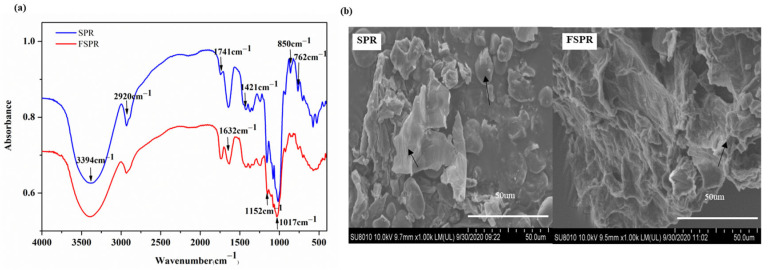
The FTIR spectrum (**a**) and scanning electron microscopy images (**b**) of sweet potato residues fermented with combined strains; SPR: Sweet potato residues; FSPR: Sweet potato residues fermented under optimal process; Arrows: the representative morphological structure of starch and dietary fiber in the images.

**Figure 3 foods-12-00128-f003:**
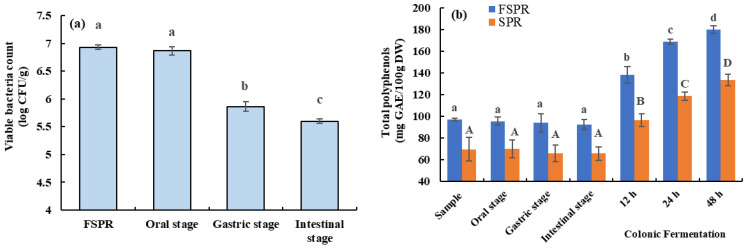
Viable bacteria counts (**a**) and total polyphenol contents (**b**) during in vitro digestion and colonic fermentation; Uppercase letters: Significantly different (*p* < 0.05) by Duncan’s test of SPR (Sweet potato residues) group; Lowercase letters: Significantly different (*p* < 0.05) by Duncan’s test of FSPR (Sweet potato residues fermented under optimal process) group.

**Figure 4 foods-12-00128-f004:**
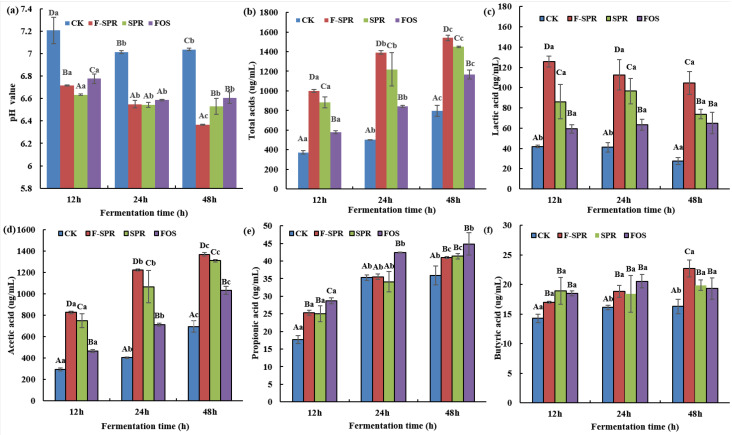
The pH value (**a**), total acids (**b**), lactic acid (**c**) and short chain fatty acids (**d**–**f**) during in vitro colonic fermentation; CK: Colonic fermentation without carbon source supplement; F-SPR: Colonic fermentation with FSPR (Sweet potato residues fermented under optimal process) supplement; SPR: Colonic fermentation with SPR (Sweet potato residues) supplement; FOS: Colonic fermentation with FOS (fructo-oligosaccharides) supplement; Uppercase letters: Significantly different (*p* < 0.05) of different samples supplement at same fermentation time; Lowercase letters: Significantly different (*p* < 0.05) of each sample at different fermentation time.

**Figure 5 foods-12-00128-f005:**
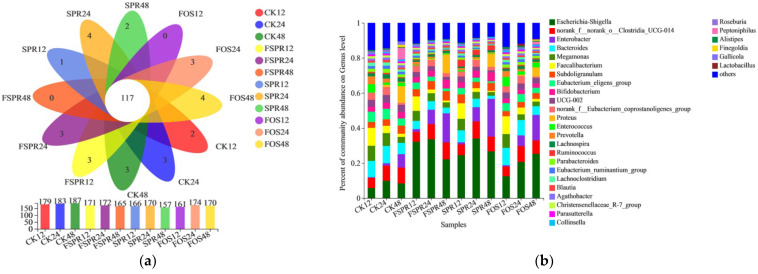
Venn diagram (**a**) and percent of community abundance (**b**) of gut microbiota at genus level during in vitro colonic fermentation; CK12, CK24, and CK48: Colonic fermentation without carbon source supplement for 12, 24, and 48 h; FSPR12, FSPR24, and FSPR48: Colonic fermentation with FSPR (Sweet potato residues fermented under optimal process) supplement for 12, 24, and 48 h; SPR12, SPR24, and SPR48: Colonic fermentation with SPR (Sweet potato residues) supplement for 12, 24, and 48 h; FOS12, FOS24, and FOS48: Colonic fermentation with FOS (fructo-oligosaccharides) supplement for 12, 24, and 48 h.

**Figure 6 foods-12-00128-f006:**
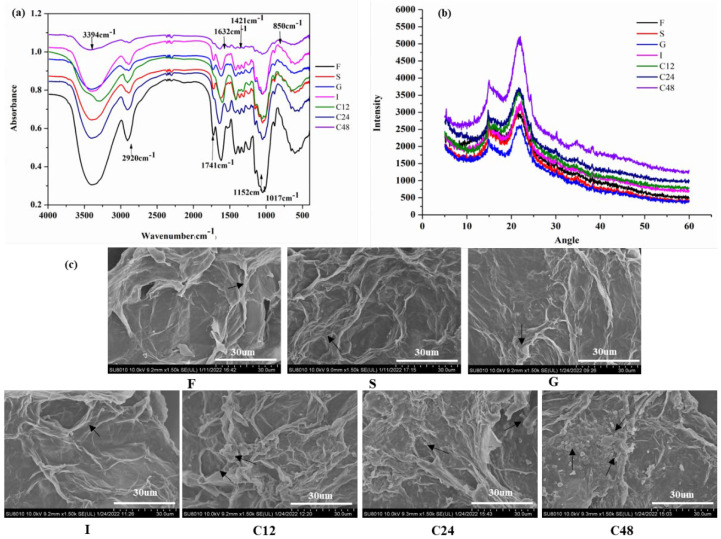
The FTIR spectrum (**a**), X-ray diffraction patterns (**b**) and scanning electron microscopy images (**c**) of total dietary fiber from sweet potato residues fermented with combined strains during in vitro digestion and colonic fermentation. F: TDF of FSPR; S: TDF of FSPR after simulated salivary di-gestion; G: TDF of FSPR after simulated gastric digestion; I: TDF of FSPR after simulated intestinal digestion; C12, C24, and C48: TDF of FSPR after simulated colonic fermentation for 12, 24, and 48 h; Arrows: the representative morphological structure of TDF in the images.

**Table 1 foods-12-00128-t001:** Nutritional composition, pH, titratable acid, organic acids, total polyphenol contents, and antioxidant activities of sweet potato residues fermented with combined strains.

Index	SPR	FSPR
Total starch	51.01 ± 2.65	3.40 ± 0.40
Reducing sugars	2.50 ± 0.13	18.02 ± 0.81
Insoluble dietary fiber	25.09 ± 3.61	20.59 ± 2.81
Soluble dietary fiber	10.17 ± 0.95	15.02 ± 0.58
Total dietary fiber	35.26 ± 2.85	35.61 ± 2.97
Protein	3.95 ± 0.09	3.88 ± 0.08
Soluble protein	475.74 ± 12.42	141.53 ± 5.76
Ash	2.41 ± 0.04	3.29 ± 0.12
pH	5.48 ± 0.12	3.26 ± 0.08
Titratable acid	1.06 ± 0.00	6.86 ± 0.63
Lactic acid	0.05 ± 0.01	58.01 ± 1.42
Acetic acid	0.10 ± 0.01	1.66 ± 0.20
Total polyphenol contents	70.47 ± 10.89	95.74 ± 1.20
ABTS	12.70 ± 0.57	13.70 ± 0.85
DPPH	33.92 ± 1.90	57.07 ± 1.24
FRAP	27.18 ± 1.36	60.51 ± 1.03

SPR: Sweet potato residues; FSPR: Sweet potato residues fermented under optimal process; Total titratable acid (mL NaOH/g DW); organic acids (mg/g DW); soluble protein (mg/100 g DW); total polyphenol contents (mg GAE/100 g DW); ABTS and DPPH (mg AAE/100 g DW); FRAP (mg TE/100 g DW); others (g/100 g DW).

**Table 2 foods-12-00128-t002:** The relative abundance of volatile components in the sweet potato residues fermented with combined strains.

Volatile Compounds	Area% of Each Sample	
	SPR	FSPR
Acids	ND	34.26±6.95
Decanedioic acid	ND	1.46 ± 0.46
Sorbic Acid	ND	28.68 ± 5.57
Acetic acid	ND	2.60 ± 0.76
Hexanoic acid	ND	1.52 ± 0.16
Aldehydes	68.08 ± 4.58	26.33 ± 1.83
Nonanal	14.63 ± 1.14	10.98 ± 1.35
Decanal	9.60 ± 1.53	6.30 ± 1.11
Benzaldehyde	4.04 ± 1.01	1.70 ± 1.32
(E)-2-Nonenal	35.61 ± 0.14	6.84 ± 0.44
Benzaldehyde, 4-(1-		
methylethyl)-	2.90 ± 0.51	0.51 ± 0.04
(E, E)-2,4-Nonadienal	1.30 ± 0.25	ND
Alcohols	ND	1.33 ± 0.26
Geraniol	ND	1.34 ± 0.12
Phenylethyl Alcohol	ND	0.56 ± 0.06
3,5-Dimethyl-Cyclohexanol	ND	0.77 ± 0.08
Esters	1.60 ± 0.24	1.88 ± 0.09
Hexadecanoic acid methyl ester	1.60 ± 0.24	1.88 ± 0.09
Ketones	1.68 ± 0.43	6.08 ± 1.09
3-Hydroxybutanone	ND	1.59 ± 0.53
2-Nonanone	1.68 ± 0.43	2.43 ± 0.05
2-Undecanone	ND	0.82 ± 0.16
Dihydro-5-pentyl-2(3H)-Furanone	ND	2.83 ± 0.35
Others	22.52 ± 3.32	19.09 ± 2.31
2-pentyl-Furan	10.16 ± 1.51	12.19 ± 0.53
Heptadecane	5.45 ± 0.3	ND
Dodecane	ND	3.08 ± 1.54
Eicosane	3.87 ± 1.05	1.55 ± 0.04
1-methyl-Naphthalene	3.04 ± 0.46	0.61 ± 0.05
Butylated Hydroxytoluene	ND	1.66 ± 0.15

SPR: Sweet potato residues; FSPR: Sweet potato residues fermented under optimal process.

**Table 3 foods-12-00128-t003:** Microbial α-diversity analysis of FSPR during in vitro colonic fermentation.

	Ace	Chao	Shannon	Simpson
CK12	390.45 ± 13.81 a	386.53 ± 17.94 b	3.99 ± 0.03 fg	0.04 ± 0.00 a
CK24	384.88 ± 8.20 bc	379.98 ± 5.83 b	4.06 ± 0.01 g	0.03 ± 0.00 a
CK48	394.31 ± 11.25 c	395.69 ± 17.81 b	3.90 ± 0.06 ef	0.04 ± 0.00 a
FSPR12	377.92 ± 8.16 bc	377.13 ± 5.66 b	3.39 ± 0.05 ef	0.12 ± 0.01 a
FSPR24	375.21 ± 20.8 bc	385.16 ± 15.19 b	3.28 ± 0.08 bc	0.13 ± 0.02 d
FSPR48	332.01 ± 43.47 ab	363.08 ± 36.37 b	3.27 ± 0.14 bc	0.09 ± 0.01 c
SPR12	387.48 ± 9.20 bc	383.50 ± 8.44 b	3.55 ± 0.15 d	0.08 ± 0.03 c
SPR24	381.06 ± 29.71 bc	382.49 ± 30.75 b	3.22 ± 0.04 b	0.13 ± 0.01 d
SPR48	320.21 ± 42.35 a	311.53 ± 40.63 a	3.01 ± 0.11 a	0.12 ± 0.02 d
FOS12	362.36 ± 9.21 abc	365.92 ± 16.49 b	3.89 ± 0.05 ef	0.05 ± 0.00 ab
FOS24	386.42 ± 20.96 bc	387.90 ± 21.59 b	3.79 ± 0.03 e	0.06 ± 0.00 b
FOS48	395.60 ± 63.37 c	394.62 ± 37.72 b	3.41 ± 0.04 c	0.10 ± 0.00 c

FSPR: Sweet potato residues fermented under optimal process; CK12, CK24, and CK48: Colonic fermentation without carbon source supplement for 12, 24, and 48 h; FSPR12, FSPR24, and FSPR48: Colonic fermentation with FSPR (Sweet potato residues fermented under optimal process) supplement for 12, 24, and 48 h; SPR12, SPR24, and SPR48: Colonic fermentation with SPR (Sweet potato residues) supplement for 12, 24, and 48 h; FOS12, FOS24, and FOS48: Colonic fermentation with FOS (fructo-oligosaccharides) supplement for 12, 24, and 48 h. Different letters indicates significant differences (*p* < 0.05).

## Data Availability

The data presented in this study are available on request from the corresponding author.

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
