# Peer review of "The Sustainability of Sweet Potato Residues from Starch Processing By-Products: Preparation with Lacticaseibacillus rhamnosus and Pediococcus pentosaceus, Characterization, and Application"

_foods, 2022, doi:10.3390/foods12010128_

Round 1

Reviewer 1 Report

Dear Editors and authors, 

The manuscript has good idea but it needs to add some corrects and  modifications.

1-The abstract of the manuscript is very weak and needs to add some important results obtained during the study. 

2-Page 4 , line 151 ,  pH value, total titratable acid (TTA) need to add reference , I suggest  (Al-Sahlany, S. T., & Niamah, A. K. (2022). Bacterial viability, antioxidant stability, antimutagenicity and sensory properties of onion types fermentation by using probiotic starter during storage. Nutrition & Food Science).

3- Page 4 line 156 , High performance anion exchange chromatography-pulsed amperometric method need to add reference , I suggest (Barros, R. F., Cutrim, C. S., Costa, M. P. D., Conte, C. A., & Cortez, M. A. S. (2019). Lactose hydrolysis and organic acids production in yogurt prepared with different onset temperatures of enzymatic action and fermentation. Ciência Animal Brasileira20.‏)

4- Page 4 line 165, Fourier-transform infrared (FTIR) spectroscopy analysis needs to add new reference.

5-Page 4 line 174, Scanning electron microscopy (SEM) analysis needs to add new reference.

6-Page 5 line 210, What was PCR program  used in Gut microbiota analysis method? And this method needs to add new reference.

7-It is wrong to abbreviate the word logarithm (lg), it should be the abbreviation Log. in all the manuscript sections.

8-Figure 1, What do the abbreviations a and A mean? Abbreviations should be explained below each figure or table in the manuscript. Standard units must be added to the y-axis such as Log CFU/ gm.

9-Figure 2. The FTIR spectrum was estimated from 4000 - 500 cm, while in the work methods it was mentioned from 4000 - 400 cm, it must be confirmed and corrected. 

10-Many of the abbreviations in Figure 4 are unclear and should be mentioned and their details below the figure, such as:CK.

11-Figure 6. The FTIR spectrum was estimated from 4000 - 500 cm, while in the work methods it was mentioned from 4000 - 400 cm, it must be confirmed and corrected. 

12-Conclusions contain many results, they must be rewritten again and any result should be excluded.

Author Response

Dear editor Dr. Jessie Xu,

Submission of the revised manuscript entitled " The sustainability of sweet potato residues from starch processing by-products: preparation with Lacticaseibacillus rhamnosus and Pediococcus pentosaceus, characterization, and application " (Manuscript ID: foods-2037440).

Your comments and those of the reviewers on the above manuscript are well appreciated. We have revised the manuscript based on the reviewers’ comments, and corrections made in the manuscript are marked up in red. You can also find the corrections made in the list attached to this letter.

Thank you for your favorable consideration.

Yours sincerely,

Hongnan Sun

Institute of Food Science and Technology, Chinese Academy of Agricultural Sciences; Key Laboratory of Agro-Products Processing, Ministry of Agriculture and Rural Affairs

Tel/ Fax: +86 10 6281 5541

Response to Reviewer#1

1-The abstract of the manuscript is very weak and needs to add some important results obtained during the study.

Reply: The important results were added in abstract according to the suggestion. Please see page 1, line 19-26 in the revised manuscript for details.

2-Page 4, line 151, pH value, total titratable acid (TTA) need to add reference, I suggest (Al-Sahlany, S. T., & Niamah, A. K. (2022). Bacterial viability, antioxidant stability, antimutagenicity and sensory properties of onion types fermentation by using probiotic starter during storage. Nutrition & Food Science).

Reply: Thank you for the kind suggestion. The related reference was added. Please see page 4, line 156; page 14, line 564-565 in the revised manuscript for details.

3- Page 4 line 156, High performance anion exchange chromatography-pulsed amperometric method need to add reference, I suggest (Barros, R. F., Cutrim, C. S., Costa, M. P. D., Conte, C. A., & Cortez, M. A. S. (2019). Lactose hydrolysis and organic acids production in yogurt prepared with different onset temperatures of enzymatic action and fermentation. Ciência Animal Brasileira20.‏)

Reply: Thank you for the kind suggestion. The equipment and mobile phase used in the related reference were HPLC system equipped with refractive index RID-10A detectors and 3 mmol/L H2SO4 solution, respectively. While High performance anion exchange chromatography-pulsed amperometric detector (HPAEC-PAD) and 0.8 mmol/L KOH solution were used in our research. Therefore, the more suitable reference was added. Please see page 4, line 158-159 in the revised manuscript for details.

4- Page 4 line 165, Fourier-transform infrared (FTIR) spectroscopy analysis needs to add new reference.

Reply: The related reference was added. Please see page 4, line 172 in the revised manuscript for details.

5-Page 4 line 174, Scanning electron microscopy (SEM) analysis needs to add new reference.

Reply: The related reference was added. Please see page 4, line 178 in the revised manuscript for details.

6-Page 5 line 210, What was PCR program used in Gut microbiota analysis method? And this method needs to add new reference.

Reply: Thank you for the kind suggestion. The PCR program and related reference were added. Please see page 5, line 220-223, page 15, line 573-574 in the revised manuscript for details.

7-It is wrong to abbreviate the word logarithm (lg), it should be the abbreviation Log. in all the manuscript sections.

Reply: We are so sorry for this mistake. The related mistakes were all revised for the whole manuscript. Please see page 6, line 252-283 in the revised manuscript for details.

8-Figure 1, What do the abbreviations a and A mean? Abbreviations should be explained below each Figure or table in the manuscript. Standard units must be added to the y-axis such as Log CFU/ g.

Reply: The abbreviations were explained below Figure 1. The standard units were also added to the y-axis. Furthermore, the abbreviations were also explained below Figure 2 and Figure 3. Please see page 6, line 269-274, page 10, line 396-397, and page 10, line 408-411 in the revised manuscript for details.

9-Figure 2. The FTIR spectrum was estimated from 4000 - 500 cm, while in the work methods it was mentioned from 4000 - 400 cm, it must be confirmed and corrected. 

Reply: The FTIR spectrum was corrected in Figure 2. Please see page 10, line 393-394 in the revised manuscript for details.

10-Many of the abbreviations in Figure 4 are unclear and should be mentioned and their details below the Figure, such as: CK.

Reply: Thank you for the kind suggestion. The abbreviations were explained below Figure 4, as well as Figure 5. Please see page 11, line 430-436, page 12, line 474-480 in the revised manuscript for details.

11-Figure 6. The FTIR spectrum was estimated from 4000 - 500 cm, while in the work methods it was mentioned from 4000 - 400 cm, it must be confirmed and corrected. 

Reply: The FTIR spectrum was corrected in Figure 6. Please see page 13, line 505-506 in the revised manuscript for details.

12-Conclusions contain many results, they must be rewritten again and any result should be excluded.

Reply: The results were excluded in the conclusion. Please see page13-14, line 513-516 in the revised manuscript for details.  

Reviewer 2 Report

Dear colleagues. First of all I want to congratulate you for this exceptional work. The study of the effects of fermentation on waste from the food industry is a very interesting aspect of current science because it can allow us to reduce the impact of our activities and we can also obtain products that can be key to maintaining our long term health.I just wanted to make some points that I think will help to understand your work, and perhaps serve to improve your exceptional work.
The first comment is about abbreviations; the abbreviation "TDF" is not defined anywhere in the manuscript. Although it could be identified as "Total Dietary Fiber", you already know that in science speculation is better located in the field of hypotheses and conclusions. It would be convenient to clarify this abbreviation where it appears for the first time, that is, on line 191 of page 4 of the manuscript. Similarly, the abbreviations "IDF" (Insoluble Dietary Fiber online 287 page 6) and SDF (Sloble Dietary Fiber online 261 page 6) should be clarified.
On the other hand, Figure 1 distinguishes between two data series that are represented in a differentiated way using the letters of the Roman alphabet from A to E (or from a to e) in lowercase or uppercase. Reading the text of the manuscript or that of the caption, it is difficult to understand the differences between the two series represented. Given that they result in key data for the proper understanding of this work, it would be interesting to include some clarification in this sense that helps to understand what it is represented.

I hope these comments can help you in some way to improve, even in this modest way, the remarkable manuscript sent to the journal.

Author Response

Dear editor Dr. Jessie Xu,

Submission of the revised manuscript entitled " The sustainability of sweet potato residues from starch processing by-products: preparation with Lacticaseibacillus rhamnosus and Pediococcus pentosaceus, characterization, and application " (Manuscript ID: foods-2037440).

Your comments and those of the reviewers on the above manuscript are well appreciated. We have revised the manuscript based on the reviewers’ comments, and corrections made in the manuscript are marked up in red. You can also find the corrections made in the list attached to this letter.

Thank you for your favorable consideration.

Yours sincerely,

Hongnan Sun

Institute of Food Science and Technology, Chinese Academy of Agricultural Sciences; Key Laboratory of Agro-Products Processing, Ministry of Agriculture and Rural Affairs

Tel/ Fax: +86 10 6281 5541

Response to Reviewer#2

Dear colleagues. First of all I want to congratulate you for this exceptional work. The study of the effects of fermentation on waste from the food industry is a very interesting aspect of current science because it can allow us to reduce the impact of our activities and we can also obtain products that can be key to maintaining our long term health. I just wanted to make some points that I think will help to understand your work, and perhaps serve to improve your exceptional work.

1-The first comment is about abbreviations; the abbreviation "TDF" is not defined anywhere in the manuscript. Although it could be identified as "Total Dietary Fiber", you already know that in science speculation is better located in the field of hypotheses and conclusions. It would be convenient to clarify this abbreviation where it appears for the first time, that is, on line 191 of page 4 of the manuscript. Similarly, the abbreviations "IDF" (Insoluble Dietary Fiber online 287 page 6) and (Sloble Dietary Fiber online 261 page 6) should be clarified.

Reply: The abbreviation "TDF", "SDF", and "IDF" were clarified where it appears for the first time. Please see page 4, line 194-195, page 6, line 268, and page 7, line 296 in the revised manuscript for details.

2-On the other hand, Figure 1 distinguishes between two data series that are represented in a differentiated way using the letters of the Roman alphabet from A to E (or from a to e) in lowercase or uppercase. Reading the text of the manuscript or that of the caption, it is difficult to understand the differences between the two series represented. Given that they result in key data for the proper understanding of this work, it would be interesting to include some clarification in this sense that helps to understand what it is represented.

I hope these comments can help you in some way to improve, even in this modest way, the remarkable manuscript sent to the journal.

Reply: The abbreviations were explained below Figure 1. Please see page 6, line 269-274 in the revised manuscript for details.

Round 2

Reviewer 1 Report

Dear Editors, 

The authors made all the appropriate changes to the manuscript to make it better, and I now suggest that it be published as is.